



# Building a cloud in the Southeast Atlantic: Understanding low-cloud controls based on satellite observations with machine learning

Julia Fuchs[1,2], Jan Cermak[1,2], and Hendrik Andersen[1,2]

[1]Institute of Meteorology and Climate Research, Karlsruhe Institute of Technology (KIT), Karlsruhe, Germany.
[2]Institute of Photogrammetry and Remote Sensing, Karlsruhe Institute of Technology (KIT), Karlsruhe, Germany.

**Correspondence:** Julia Fuchs (julia.fuchs@kit.edu)

**Abstract.** Understanding the processes that determine low-cloud properties and aerosol-cloud interactions (ACI) is crucial for the estimation of their radiative effects. However, the covariation of meteorology and aerosols complicates the determination of cloud-relevant influences and the quantification of the aerosol-cloud relation.

This study identifies and analyzes sensitivities of cloud fraction and cloud droplet effective radius to their meteorological

and aerosol environment in the atmospherically stable Southeast Atlantic during the biomass-burning season. The effect of geophysical parameters on clouds is investigated based on a machine learning technique, gradient boosting regression trees (GBRTs), using a combination of satellite and reanalysis data as well as trajectory modeling of air-mass origins. A comprehensive, multivariate analysis of important drivers of cloud occurrence and properties is performed and evaluated.

The statistical model reveals marked subregional differences of relevant drivers and processes determining low clouds in the

Southeast Atlantic. Cloud fraction is sensitive to changes of lower tropospheric stability in the oceanic, southwestern subregion, while in the northeastern subregion it is governed mostly by surface winds. In the pristine, oceanic subregion large-scale dynamics and aerosols seem to be more important for changes of cloud droplet effective radius than in the polluted, near-shore subregion, where free tropospheric temperature is more relevant. This study suggests the necessity to consider distinct ACI regimes in cloud studies in the Southeast Atlantic.

# 1   Introduction

Low-level clouds play a major role in the climate system via their impact on the Earth's energy budget and water cycle (Boucher et al., 2013). However, the estimation of their potentially large negative radiative effect is prone to large uncertainties as processes that govern cloud micro- and macro-physical properties, i.a. aerosol-cloud interactions (ACI), and the impact of changing environmental conditions on low clouds, are not sufficiently understood (Bony and Dufresne, 2005; Medeiros et al.,

2008). Maritime stratocumulus clouds, persisting over the relatively clean southern oceans are thought to be especially sensitive to aerosols, exerting a strong cloud albedo effect of -0.2 W m$^{-2}$ (Platnick and Twomey, 1994; Quaas et al., 2008). One of these regions, the Southeast Atlantic (SEA), has become a very popular region for studies of low-cloud processes and ACI in the last decade (e.g., Chand et al., 2009; Muhlbauer et al., 2014; Painemal et al., 2014; Andersen and Cermak, 2015; Adebiyi et al., 2015; Fuchs et al., 2017).



The semi-permanent low-cloud cover of the SEA is driven by the cold Benguela current offshore the Namibian/Angolan coast and maintained by large-scale subsidence (Wood, 2012). During the biomass-burning season in July-August-September (JAS), carbonaceous aerosols are advected over the oceanic boundary layer and frequently build a thick layer above the clouds. Black carbon aerosol particles can act as cloud condensation nuclei as they are entrained at cloud-top (Seinfeld et al., 2016) or indirectly alter cloud cover through the strengthening of the inversion by absorption of shortwave radiation above the cloud (Wilcox, 2010; Bond et al., 2013; Li et al., 2013).

Despite advances on the basis of large eddy simulations (e.g., Yamaguchi and Randall, 2008; Jones et al., 2014), lagrangian approaches (e.g., Mauger and Norris, 2010) and observational studies (e.g., Zuidema et al., 2016), the complex mechanisms between low clouds, boundary layer processes, thermodynamics and large-scale circulation are not sufficiently understood. Untangling the drivers of cloud properties is challenging, as meteorological parameters and aerosols covary (Mauger and Norris, 2007; Fan et al., 2016), vary spatially and have different time scales (Jones et al., 2014; Eastman et al., 2016; de Szoeke et al., 2016).

In a recent study, Fuchs et al. (2017) showed that air-mass origins can explain some of the variability of cloud microphysics in the SEA, with clear spatial differences in the involved processes. Analyses of cloud sensitivities in the SEA would therefore benefit from a subregional determination of large-scale, thermodynamic and aerosol drivers of cloud property changes. Relevant mechanisms for changes of low-cloud properties are studied here focusing on two questions:

- What are the subregional differences of cloud sensitivities to various geophysical parameters?

- How do these determinants influence cloud properties and their response to atmospheric aerosol loading?

In this study a machine learning approach is used to predict cloud fraction and cloud droplet effective radius in the SEA based on satellite and reanalysis data. This study does not aim to simulate microphysical cloud processes and individual feedback mechanisms at the level of detail of a cloud-resolving model, but instead intends to represent non-linear patterns of cloud adjustments to the large-scale and thermodynamic environment in a coherent, multivariate statistical model.

## 2 Methods

### 2.1 Data

Cloud fraction (CF), cloud droplet effective radius (REF) and aerosol optical depth (AOD) are obtained from the 8-day level 3 (L3) product of the MODerate-resolution Imaging Spectroradiometer (MODIS) instrument aboard the Aqua platform (collection 6). The data cover a temporal range from 2002 to 2012 during the biomass-burning season in July-August-September. The REF product is based on single-layer liquid clouds to avoid effects of overlapping cirrus clouds (Hubanks et al., 2018). The following thermodynamic and dynamic parameters of the ERA-Interim reanalysis dataset of the European Centre for Medium-Range Weather Forecasts (ECMWF) are used: lower tropospheric stability (LTS), relative humidity at 950/850/700 hPa (RH950, RH850, RH700), surface wind speed at 10 m (WSP10), sea surface temperature (SST) and temperature at 700





hPa (T700), zonal wind speeds at 600 hPa (U600) and mean sea level pressure (MSLP).

The ERA-Interim reanalysis data is also used in the calculation of 5-days backward air-mass trajectories with the HYSPLIT

model using geopotential height, relative humidity, temperature, u/v wind components, vertical velocity at 37 pressure levels.

The backward trajectories are initialized at 12 UTC, at each grid point of the study area and at a subregional mean cloud-top

altitude obtained from the CALIPSO Level-2 5 km layer cloud product (version 3, daytime).

All meteorological variables are resampled from 0.5 degrees to the MODIS L3 resolution of 1 degrees and temporally aver-

aged to the corresponding 8-day product. The temporal resolution of 8 days allows to combine large-scale and thermodynamic

forcings of cloud properties, as clouds adjust on different time scales (hours to several days) to the large-scale circulation and

their thermodynamic environment (Klein, 1997; McCoy et al., 2017; Adebiyi and Zuidema, 2018).

## 2.2 Subregional GBRT models

In this study CF and REF are simulated based on a selected predictor set (AOD and meteorological parameters) in the SEA

(10°–20° S, 0°–10° E) using Gradient Boosting Regression Trees (GBRTs). To account for subregional spatial variability of

e.g. cloud altitude, aerosol occurrence, boundary layer dynamics and large-scale dynamics, the study area is divided into four

equal-sized subregions of 5° by 5°: the northwestern (NW), northeastern (NE), southwestern (SW), southeastern (SE) subre-

gion. Consequently, drivers of CF and REF are analyzed in the environmental context of each subregion individually, yielding

eight subregional statistical models (four subregions x two predictands).

GBRTs are a highly robust machine learning technique aimed at mapping the relationship between a set of predictors and a

predictand. The GBRT algorithm produces an ensemble of many weak prediction models ("base learners" or trees), which are

expanded in stages, following the gradient descent of a specified loss function (Friedman, 2001; Natekin and Knoll, 2013).

These statistical models are widely used in environmental and atmospheric sciences (e.g., Sayegh et al., 2016; Carslaw and

Taylor, 2009) due to their predictive power, simple implementation and flexibility toward qualitative and quantitative data

(Hastie et al., 2009). However, GBRTs require careful parameter tuning (e.g. boosting iterations, learning rate), as the goal is to

represent the given data and relationships as accurately as possible, without overfitting the model. The GBRT implementation

of the scikit-learn library was used and adapted to this end (Pedregosa et al., 2011).

To train, test and validate the statistical models, the data set is split into three random parts, the training (50 %), test (20 %)

and validation (30 %) data sets. The model setup is tuned based on the training data by testing various scenarios specified

by a parameter grid through 3-fold cross-validated search. During cross-validation, the training set is divided into three parts:

two thirds are used for training and one third for testing. Each parameter combination from the grids, listed in Table 1, is

evaluated based on the $r^2$-score obtained in correlating predicted and observed output. The obtained hyper-parameter with the

highest performance is chosen to set up the model. In general, a high number of boosting iterations and a low learning rate

will increase the model's ability to generalize, its performance and computational demand. The Huber loss function is chosen

due to its higher robustness compared to other continuous loss functions, e.g. least squares (Huber, 1964; Natekin and Knoll,

2013). A subsample rate (a random fraction of the training data used for fitting) of 0.8 is selected to reduce variance and in-

crease model robustness. All remaining parameter settings are left at their default values as provided by the gradient boosting





**Table 1.** Model parameter grid tested during 3-fold cross validation.

| model parameter | impact on model performance | parameter grid tested |
|---|---|---|
| learning rate | low values allow for better generalization | [0.05,0.04,0.03,0.02,0.01,0.009,0.007,0.003] |
| boosting iterations | large values improve performance, but risk overfitting | [2000,2400,2800,3200,3600,4000] |
| maximum depth of a tree | small numbers prevent overfitting | [2,3,4] |
| minimum samples per leaf | small values risk overfitting | [10,14,18] |

regressor function (Pedregosa et al., 2011).

Providing the optimal model setup, the model is fitted to the training data. In parallel, the test data set is used to regularize the

GBRTs by determining the final boosting iteration. The learning stops when the mean squared error (MSE) of the test data set is increasing or constant five times in a row.

To evaluate the overall performance of the GBRTs, two measures, the coefficient of determination ($r^2$) and the root mean squared error (RMSE) between predicted and observed CF (REF) are calculated using the independent validation dataset. To ensure comparability between the RMSE of the CF and REF performance the RMSE is normalized (NRMSE) by the difference

between the maximum and minimum observed values.

The final model can be interpreted using 'partial dependence', which expresses the averaged change of a cloud property relative to a selected predictor set by averaging over all complement predictors (Friedman, 2001). This is done by computing an average prediction function for a given range of values ($1^{st}$–$99^{th}$ percentile) estimated from the target predictor. Each grid point of the target predictor is fixed while the values of the complement predictors vary over their marginal probability density. As

a result, the partial dependence represents the influence of one target variable, accounting for the full meteorological variation of the complement predictors. The absolute difference of the maximum and minimum partial dependence is further used to compare the cloud property response due to the different predictors, thus to obtain a general measure for the most important drivers in the different subregions. In order to analyze the joined influence of two variables on the predictand, two-variable partial dependence plots are used. For regression trees the implementation of partial dependence is straightforward and can be

derived from the tree structure itself through a weighted tree traversal proposed by Friedman (2001). The partial dependence obtained from the GBRT model is added by the cloud property mean value for reference.

In general, GBRTs are a powerful tool for representing non-linear dependencies and emphasize subregionally important determinants for low clouds in the SEA. However, for the interpretation it must be considered that partial dependencies rely on a statistical model. That means that associations between predictors and predictand are not necessarily causal, as in every

statistical model.

## 2.3   Predictor selection

The predictor selection pursues the goal of creating a simple model capable of capturing general thermodynamic, dynamic, stratification and aerosol patterns relevant for changes of cloud properties and is based on findings of previous studies (e.g.,



Norris and Iacobellis, 2005; Lacagnina and Selten, 2013; McCoy et al., 2017; Andersen et al., 2017; Fuchs et al., 2017; Adebiyi and Zuidema, 2018). Twelve predictors (see Table 2 for an overview) are chosen as inputs to the GBRTs due to their known

forcing on CF and REF in the SEA. The listed parameters describe cloud-relevant environmental conditions at the sea surface (e.g., SST, MSLP), cloud level (RH950, RH850) and the free troposphere (e.g., T700, RH700).

**Table 2.** Predictors and abbreviations used in the GBRT models.

| Thermodynamic properties | Dynamic properties |
|---|---|
| Lower tropospheric stability (LTS) | Source latitude of air mass (Lat_src) |
| Sea surface temperature (SST) | Source longitude of air mass (Lon_src) |
| Temperature at 700 hPa (T700) | Wind speed at 10m (WSP) |
| Relative humidity at 700 hPa (RH700), | Zonal wind speeds at 600 hPa (U600) |
| at 850 hPa (RH850), at 950 hPa (RH950) | Mean sea level pressure (MSLP) |
| **Aerosol property** | |
| Aerosol optical depth (AOD) | |

The lower tropospheric stability, a proxy for inversion strength, and sea surface temperature are primary controls for the multi-day and seasonal cloud occurrence in the SEA (Klein and Hartmann, 1993; Klein et al., 1995; de Szoeke et al., 2016). Here,

LTS is defined as the difference between potential temperature ($\theta$) at 850 hPa and 1000 hPa as described in Painemal and Zuidema (2010).

As relative humidity is essential for cloud formation processes and characteristics in the free troposphere and at cloud level (Wood, 2012; Jones et al., 2014; Bretherton et al., 2013; Andersen et al., 2017), free tropospheric humidity at 700 hPa and at two different cloud levels (950 and 850 hPa) are selected as predictor.

The large-scale circulation and the history of air masses drive boundary layer cloudiness (Klein et al., 1995; Mauger and Norris, 2007; Fuchs et al., 2017). In order to represent the influence of external dynamics on the local cloud field, the latitude and longitude of the origin of 5-day backward trajectories (Lat_src, Lon_src) are included as predictors in the statistical models. The backward trajectories are initiated at the mean cloud-top altitude in every subregion: 1090 m (NW), 1060 m (NE), 1180 m (SW), and 810 m (SE). Air-mass dynamics, including the surface wind speed and the strength of subtropical anticyclones,

are important drivers for cloud amount, physical and radiative properties (Klein et al., 1995; Brueck et al., 2015; Kazil et al., 2015; Bretherton et al., 2013) and considered as predictors in the GBRT models. The strength of the South African Easterly Jet is observed to influence the marine boundary layer during the month of September to October through changes in stability and subsidence. It is defined as easterly wind speeds exceeding 6 m s$^{-1}$ between 5° S to 15° S at 650–600 hPa (Adebiyi and Zuidema, 2016). In this study its influence is assumed to extend over the study area and thus the zonal wind field at 600 hPa is used.

Aerosols interact with liquid clouds in a multifaceted way (Fan et al., 2016). According to Twomey's theory of the first





aerosol indirect effect, aerosols act as cloud condensation nuclei and influence cloud microphysics and albedo (Twomey, 1974). The Albrecht hypothesis states that this effect may result in a prolonged cloud lifetime, increased cloud optical thickness,

liquid water path and cloud fraction through the suppression of precipitation (Albrecht, 1989). For the investigation of cloud susceptibility to aerosols, the AOD is considered as a proxy for cloud condensation nuclei. While the aerosol index may be a better proxy for cloud condensation nuclei than AOD (Stier, 2016), its computation requires the Ångström exponent, which is not available in the 8-day MODIS L3 product (Levy et al., 2013). Studies that observed the bivariate relations between AOD and cloud properties are numerous (e.g., Kaufman, 2005, 2006; Grandey et al., 2013), but spurious correlations exist. The strength

of the relation between AOD and CF or REF is depending on satellite artifacts in the vicinity of clouds, e.g. cloud contamination and three-dimensional radiative effects (Várnai et al., 2013; Christensen et al., 2017) as well as on meteorological conditions, e.g. aerosol hygroscopic swelling with humidity (Kaufman et al., 2005; Quaas et al., 2010). In turn aerosols may alter the cloud's thermodynamic environment, through the semi-direct effect, where absorbing aerosol layers increase stability through local heating (Johnson et al., 2004; Li et al., 2013).

The choice of predictors reflects the compromise between characterizing the atmospheric state sufficiently to derive subregional patterns of relevant low-cloud drivers, without creating a model which fully covers the interactions between clouds and their environmental conditions, but lacks interpretability. The application of the GBRTs aims at finding important meteorological controls for changes in CF and REF in the subregions of the SEA.

## 3 Results and discussion

### 3.1 Validating GBRT models

In this section the statistical models are evaluated, important features within the models are identified and subsequently, partial dependencies (see Sect. 2.2 for more information) of the most important determinants are presented.

Figure 1 shows the validation results for the GBRTs predicting CF and REF in the different subregions. The performance is compared to a multiple linear regression analysis, using the same data basis. The correlation ($r^2$) of predicted and observed

values in the GBRT model ranges from 0.57 to 0.79 in the different subregional models and is clearly superior to the $r^2$ of the multiple linear regression model ranging from 0.32 to 0.58 in the different subregions. The $r^2$ range (error bars) of 10 random GBRT simulations based on 10 different data random splits typically does not exceed the $r^2$ range of the linear regression using the 10 different data random splits, indicating constant model performances. Both models show a low normalized RMSE (NRMSE), that is on average ~5 % for the GBRTs and ~7 % for the linear regression.

Considering the GBRT models only, two aspects can be noted. First, in the northern subregions the REF models perform slightly better than the CF models, and second, the CF model shows subregional variations. Differences of model skills might be attributed to a higher variability of the cloud properties and meteorological conditions prevailing in the SW compared to the NE (Fuchs et al., 2017; Adebiyi and Zuidema, 2018; Rahn and Garreaud, 2010), or point to missing information in the predictor set of the NE-CF model.





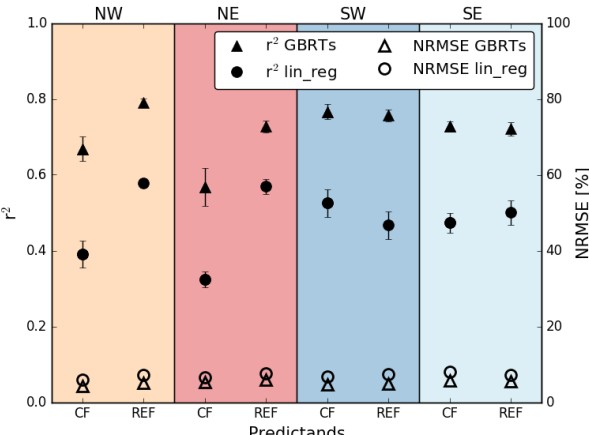

**Figure 1.** The overall mean quality of the GBRT models (triangles) is compared to a simple least squares linear regression (circles) for CF and REF in the four subregions NW, NE, SW and SE. The models are evaluated based on the coefficient of determination ($r^2$) and normalized root mean squared error (NRMSE) between predicted and observed CF (REF). The error bars range from the minimum to maximum $r^2$ obtained from 10 different models using randomly chosen training data.

As all GBRT models have been shown to adequately represent parameter relations, the statistical relationships within the models are subsequently analyzed with the purpose of inferring process relationships.

### 3.2 Sensitivity of cloud fraction and droplet radius

5   Figure 2 shows the multi-model mean absolute difference of the maximum and minimum partial dependence of CF (a) and REF (b) on the predictors as a measure for the sensitivity of these cloud properties to the various predictors.

In general, LTS, surface wind speed and relative humidity at 950 hPa play an important role for the determination of CF, however, marked subregional differences in their sensitivities can be identified (Fig. 2a). It is notable that LTS is most sensitive to CF in the southern subregions. In the northeast, the impact of relative humidity at 950 hPa on CF is markedly reduced. Here, 10   surface wind speed seems to be a key driver of CF. Changes in AOD seem to have a marked impact on CF only in the eastern subregions that are frequently exposed to high aerosol loadings.

The REF (Fig. 2b) is largely controlled by the free tropospheric temperature in the NE subregion. Here, REF is, similar to CF, strongly influenced by surface winds. In the SE, relative humidity at 950 hPa is an important driver for REF, while in the other subregions, relative humidity at 850 hPa has a stronger impact on REF due to the higher cloud level. In the SE, 15   which is regularly exposed to the continuous warm and dry air advection from the coastal and continental region, an occasional moistening through dynamical changes may have a strong effect on cloud droplets of a thin cloud layer (Adebiyi et al., 2015).

The influence of dynamical parameters such as zonal wind at 600 hPa and air-mass origin (Lon_src) on REF is especially relevant in the SW, while LTS is a prominent influence in the NW. As expected, the contribution of aerosols to changes in CF





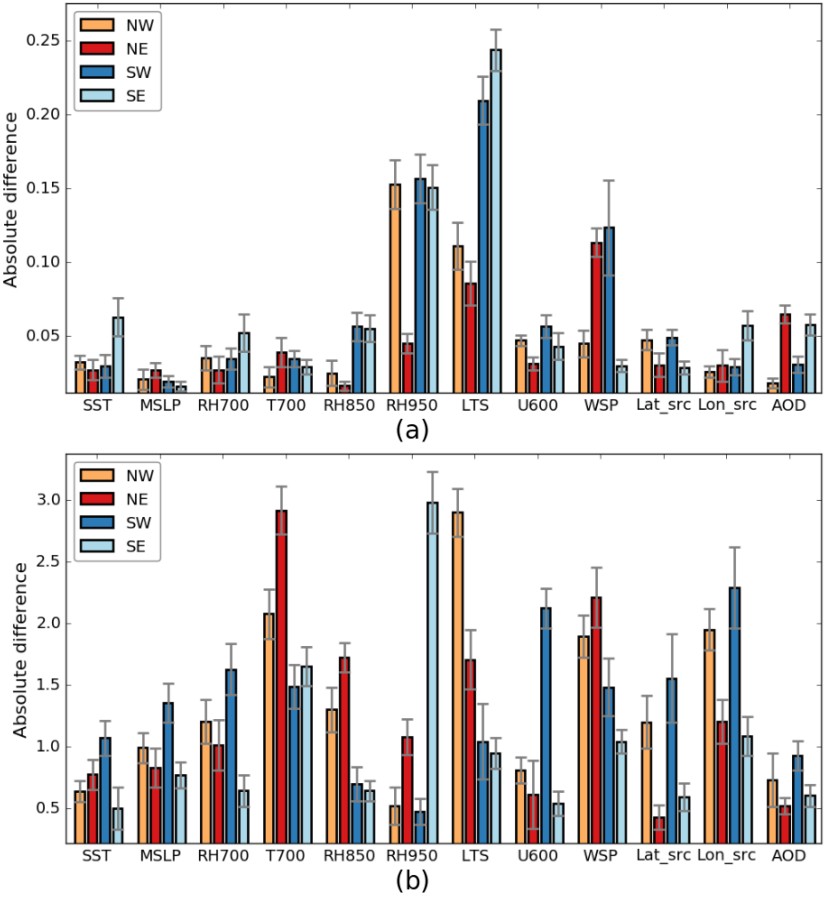

**Figure 2.** Mean absolute difference of maximum and minimum partial dependence of CF (a) and REF (b) on the predictors in the four subregions (colors). 'Error' bars show the minimum and maximum absolute difference of partial dependencies of all model runs.

and REF is small compared to the main meteorological drivers. However, the absolute differences indicate that aerosols appear to be most important for REF in the relatively pristine SW.

Based on these outcomes important predictors are brought into focus and the GBRT partial dependencies of CF and REF on
5    selected predictor variables are analyzed in more detail in the following subsections.

### 3.2.1 Thermodynamics

In accordance with findings of earlier studies (Klein and Hartmann, 1993; Zhang et al., 2009), Fig. 3(a) shows that CF increases with LTS in all subregions. This relation is explained by reduced dry-air entrainment under stable conditions building a shallow,
10   well-mixed and humid cloud layer (cf. Wood and Bretherton, 2006; Wood, 2012; Myers and Norris, 2013). Under very stable conditions, above 30 K temperature difference, the sensitivity of CF to LTS seems to be saturated and further stabilization




does not increase the cloudiness anymore. This relates well to findings by Zhang et al. (2009), who detected the strongest CF sensitivity at intermediate LTS. It is remarkable that CF sensitivity to LTS in southern subregions is about twice as strong as in the northern subregions. This observation might be attributed to cloud break-up linked to midlatitude cyclones (Toniazzo et al.,

5  2011; Fuchs et al., 2017). In contrast, in the NE the impact of LTS on CF is relatively weak as this area is characterized by more stable conditions with less thermodynamic variability.

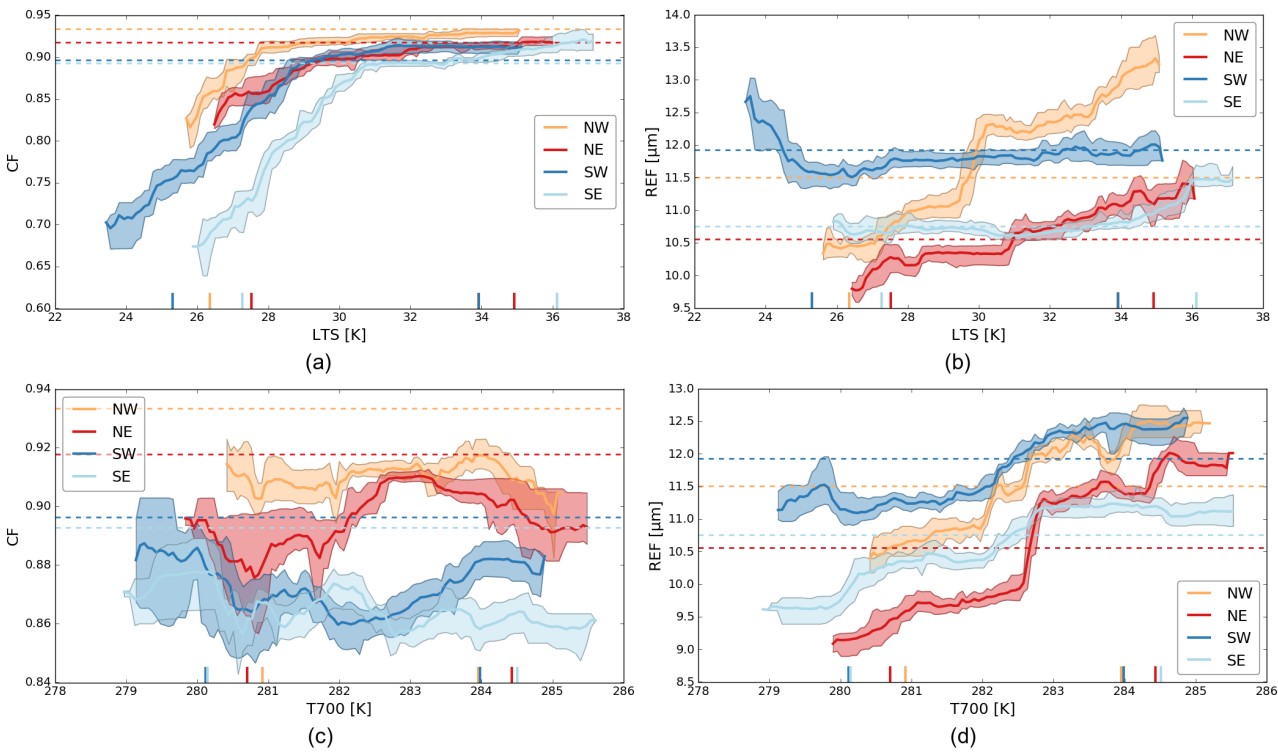

**Figure 3.** Mean partial dependence of CF and REF on LTS (a,b) and T700 (c,d) in the four subregions (colors). Shaded areas mark minimum and maximum partial dependence obtained from all model runs. Horizontal dashed lines show the predicted mean. Vertical tick marks on the x-axis indicate $5th$ and $95th$ percentile of the observations.

The relation of REF and LTS (Fig. 3b) is the strongest in the NW. A marked jump at ∼30 K may indicate the transition from a stable, relatively well-mixed coupled stratocumulus regime with larger droplets to an unstable, decoupled regime, where cloud

10  liquid water evaporation due to dry and warm air entrainment can reduce droplet size (Bretherton and Wyant, 1997).

While the partial dependence of T700 on CF shows no distinct pattern in any subregion, a strong REF sensitivity to T700 can be noticed, in particular in the NE. As droplet size is retrieved at the cloud top, it might be more sensitive to a free tropospheric warming at 700 hPa and reduced dry-air entrainment above. The cloud cover, through the cloud's vertical extent, is probably



more sensitive to the 850 hPa temperature, which is part of the LTS calculation (cf. Sect. 2.3).

### 3.2.2 Dynamics

Large-scale dynamics, here the origins of air masses, can influence cloud cover in the SEA in different ways (cf. Fuchs et al.,
2017). Figure 4(a) shows the response of CF to changes in the latitudinal origin of air masses (Lat_src). While in the eastern
subregions, CF seems largely insensitive to changes in Lat_src, CF in the western subregions is negatively associated with
Lat_src: i.e. CF decreases the further north the air-mass origin. This likely points to findings of Fuchs et al. (2017), who found
that long-distance air masses, induced by westerly disturbances, are related to increased boundary layer height, cloud fraction
and cloud droplet sizes in the western parts of the SEA. Air masses originating from ∼20° S (SW) and ∼15° S (NW) may
contribute to the reduction of CF by subsiding dry air. The shift of the CF minima between southern and northern subregions
may be interpreted as time lag of these air-mass paths, reaching the southern subregions earlier. In parallel to CF, the REF
sensitivity to the latitudinal air-mass origin is particularly strong in the western subregions of the study area, especially the SW
(Fig. 4b). The subregional difference between western and eastern subregions is even stronger than for CF. The NE shows only
a weak response of REF to the latitudinal component of the air-mass origin due to the influence of mainly continental air-mass
origins (Fuchs et al., 2017) ranging much more on the longitudinal scale (Fig. 3b).

The CF sensitivity to the surface wind field is shown in Fig. 4(c). A clear increase of CF with higher surface wind speeds can
be observed in the SW, where a change of wind speed of $1 \text{ m s}^{-1}$ entails an increase of CF of more than 10 %. Strong surface
winds may be associated with increased cold air advection and surface heat fluxes, favoring higher low-cloud amounts (Klein,
1997; Brueck et al., 2015). In all subregions, REF increases with wind speed (Fig. 4d), likely due to dynamic droplet growth
in a more turbulent boundary layer. The partial dependence of CF on the zonal wind field at 600 hPa shows a decrease in
the southern subregions, when strong westerly winds are prevailing, and may indicate cloud-free areas in more convectively-
driven systems (Fig. 4e). Weak tendencies of a CF enhancement in the southern subregions and a CF decrease in the NW due
to stronger easterly winds are apparent and may indicate the influence of the South African Easterly Jet as discussed in Adebiyi
and Zuidema (2016). As shown in Fig. 4(f), REF is largely insensitive towards the zonal wind fields at 600 hPa, presenting a
strong effect only in the SW, where westerly winds are associated with larger droplets. These characteristics may support the
effect of westerly disturbances which are more frequent in the SW.

Figure 5 shows the two-variable partial dependencies of REF on latitudinal and longitudinal air-mass origins for all four sub-
regions, underlining regional differences in the susceptibility of REF to large-scale dynamical changes. In the SW, air masses
originating from the far SW are connected to larger REF than air masses from the NE (Fig. 5c). In contrast, in the NE, larger
REF are attributed to more humid air masses from the west (Fig. 5b), while easterly and probably drier winds from the conti-
nent favor smaller REF. The origin of air masses is more important for droplet size in the SW than in the NE through its higher
subregional variability as a result of the occasional propagation of westerly disturbances.





**Figure 4.** Mean partial dependence of CF and REF on source latitude of air mass (a,b), surface wind speed (c,d) and zonal winds at 600 hPa (e,f) in the four subregions (colors). Details as in Fig. 3.

### 3.2.3 Conditions of aerosol-cloud interactions

Although the impact of aerosols on cloud properties, tends to be relatively weak, characteristic patterns are obtained in the different subregions. CF increases with AOD in all subregions, especially in the southern subregions, as shown in Fig. 6(a). This relation is found in many studies and can have both, artificial and physical reasons (e.g., Mauger and Norris, 2007; Gryspeerdt et al., 2016; Andersen et al., 2017; Adebiyi and Zuidema, 2018). The observed relation may be physically induced through the




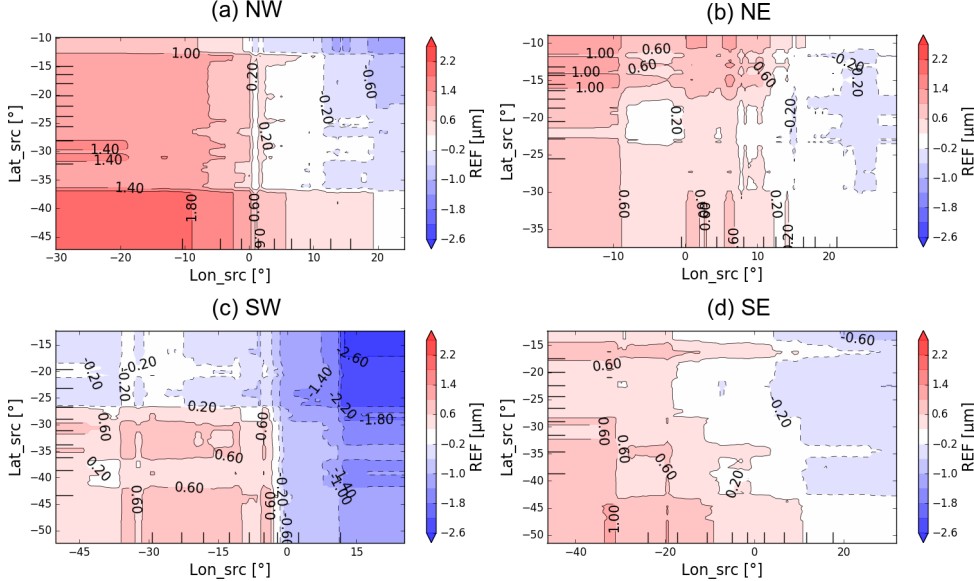

**Figure 5.** Two-variable partial dependence of REF on Lon_src and Lat_src in the in the four subregions ((a) NW, (b) NE, (c) SW, (d) SE). Solid (dashed) contour lines indicate positive (negative) deviation of the predicted mean. The tick marks on the x-axis and y-axis indicate the deciles of the observations. For this illustration only one model run selected at random was used.

availability of CCNs, increasing cloud lifetime and fractional cloudiness as aerosols are present (cf. Albrecht, 1989). It may be further explained by semi-direct effects, where absorbing carbonaceous aerosol layers heat the free troposphere causing a stabilization of the atmosphere that promotes the humidification of the cloud layer (cf. Li et al., 2013). Whether stability is enhanced by absorbing aerosols or is connected to the transport of aerosol-loaded warm air cannot be answered at this point. The effect of AOD enhancement on the AOD-CF relation due to hygroscopic swelling (Quaas et al., 2010) and wind-induced sea spray (Engström and Ekman, 2010), is thought to play a minor role due to the explicit consideration of relative humidity and surface wind speed in the statistical models. In the NE, the reason for the strong AOD-CF relation ($< 5th$ percentile of AOD) is intriguing but it is unclear to what extent it is caused by aerosol-related physical processes. It should be noted that these conditions only rarely occur.

The partial dependence of REF on the aerosol loading is shown in Fig. 6(b). The southern subregions show a comparable pattern of a REF decrease up to AOD values of ∼0.2. A subsequent REF increase up to an AOD of ∼1 can be noticed in all subregions. The response of REF at lower AOD values is especially marked in the SW. Here, a different aerosol regime (composition and size: i.e. sea salt in the SW vs. biomass burning in the NE), giant cloud condensation nuclei, larger droplets in more turbulent conditions and the closer vicinity of aerosol and cloud layers may favor stronger aerosol indirect effects (cf. Andreae and Rosenfeld, 2008; Costantino and Bréon, 2012; Painemal et al., 2014; Andersen and Cermak, 2015). Stronger aerosol effects at low aerosol loadings were also found by Andersen et al. (2016) at a global scale. These results point to a saturation of the aerosol indirect effect under highly polluted conditions, where the influence of stability may be stronger. To



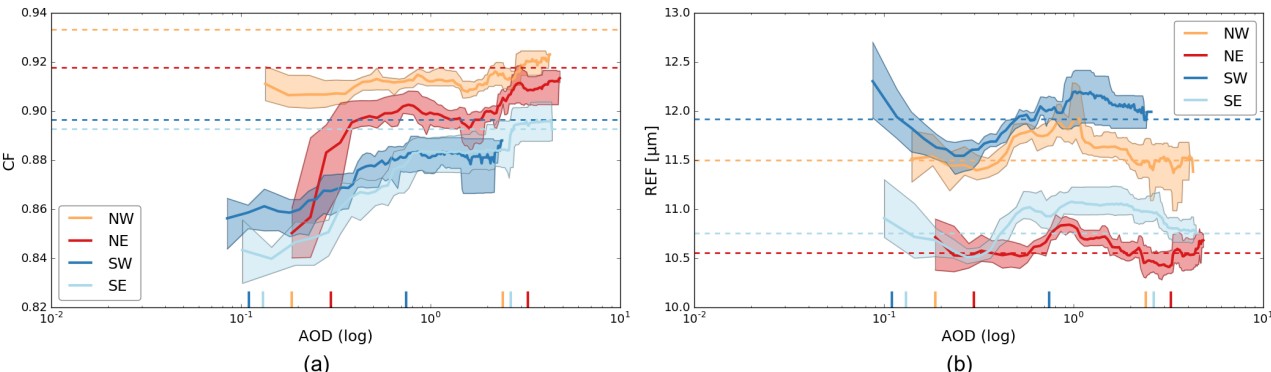

**Figure 6.** Mean partial dependence of CF (a) and REF (b) on AOD in the four subregions (colors). Description as in Fig. 3.

what extent the relationship between REF and the AOD can be attributed to an absorbing aerosol bias in the satellite retrievals (Haywood et al., 2004) or physical processes cannot be answered definitively. However, the observed subregional differences of the polluted NE versus the more pristine SW make aerosol indirect effects more likely than retrieval issues.

The two-variable partial dependencies, presented in Fig. 7 to 11, show how the sensitivities of CF and REF to aerosol loading may vary under different meteorological conditions, i.e. LTS and relative humidity at 950 and 850 hPa. All subregions of the

SEA are characterized by a stronger CF (Fig. 7) and REF sensitivity (Fig. 8) to LTS compared to AOD. In the southern subregions, CF is increased under stable and strongly polluted conditions. Here, the increase of CF with AOD is more pronounced in stable conditions, presumably due to reduced dry-air entrainment (cf. Chen et al., 2014), while CF seems to be less sensitive to aerosols in unstable conditions, where primarily low CF may result from cloud breakups in midlatitude cyclones (cf. Toniazzo et al., 2011). In contrast, a generally higher REF sensitivity to aerosols characterizes the SW. In this subregion, larger droplets

may more effectively persist, grow, and are thus susceptible to aerosols in both, stable and unstable (mixing of aerosols into the cloud layer) conditions (cf. Painemal et al., 2014). In the NE, it can further be observed that the CF sensitivity to aerosols is favored at low aerosol loading, which might be explained by the saturation of aerosol effects at higher loading (cf. de Szoeke et al., 2016).

The relation of CF (REF), humidity at 950 hPa and AOD is shown in Fig. 9 (10). Humidity at 950 hPa dominates all subre-

gions, particularly the SE, while the impact of aerosols is relatively small. In the southern subregions, though, CF increases under humid and polluted conditions (Fig. 9c,d). CF is especially sensitive to an increase of aerosol loading below a cloud level-humidity of ∼80 %, while above this level aerosol swelling is more likely to affect the AOD retrieval (cf. Adebiyi and Zuidema, 2018). As shown for CF, relative humidity is essentially related to REF, and a reduction of REF due to aerosols is apparent throughout the different humidity ranges at 850 and 950 hPa (Fig. 10 and 11). In the SW (Fig. 11c), REF may be

sensitive in drier as well as more humid conditions: while humid conditions provide larger droplets, entrainment induced by aerosols may more effectively reduce droplet size in dry conditions (cf. Chen et al., 2014).

In sum, the presented results show the potential of observing ACI susceptibilities in different thermodynamic conditions. Nev-





ertheless, the presented link between meteorological conditions and aerosol effect on clouds (indirect and semi-direct) is not necessarily causal and further effects due to aerosol processing near clouds and satellite artifacts (Sect. 2.3) may contribute to the observed cloud sensitivities.

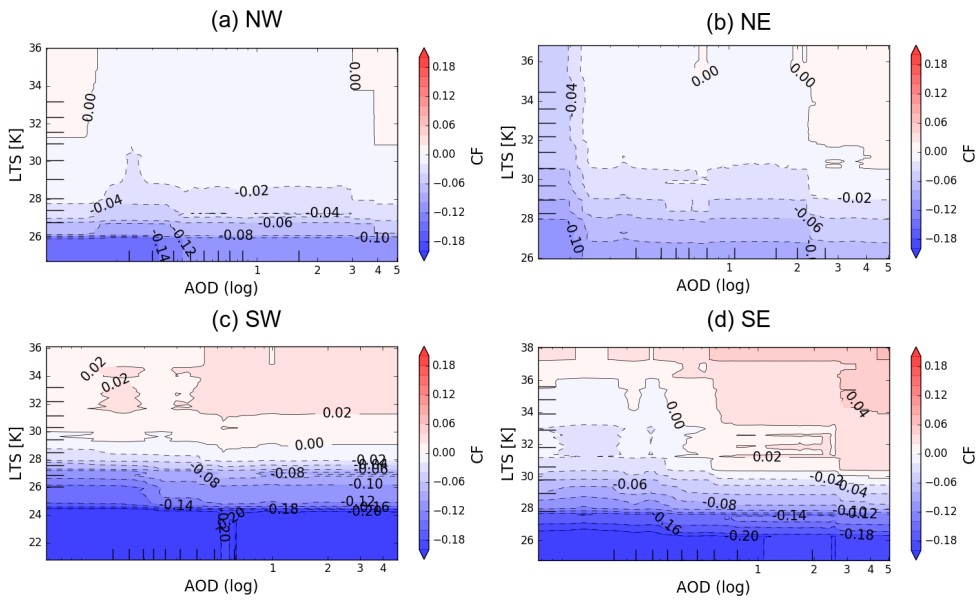

**Figure 7.** Two-variable partial dependence of CF on LTS and AOD in the four subregions NW (a), NE (b), SW (c), SE (d). Description as in Fig. 5.

## 4 Conclusions

5 In this study relevant mechanisms for changes in CF and REF are analyzed by using a GBRT model in four subregions of the Southeast Atlantic. The GBRT models perform significantly better than multiple regression analyses based on the same data (average $r^2$ of 0.72 vs. 0.48, respectively). This indicates that the GBRT models can be used to adequately represent the interactions governing the cloud system, while the methodical approach proves advantageous. The model skill varies with subregion and cloud property and features different sensitivities to the same predictor set. Outcomes of the GBRTs provide useful

10 insights of important determinants for cloud properties. By accounting for meteorological conditions and aerosol loadings the models can help untangling the various cloud processes and cloud sensitivities to aerosols in the subregions of the SEA. The subregional importance and patterns of cloud drivers and ACI sensitivities is plausible and in accordance with findings of related studies (e.g., Chen et al., 2014; McCoy et al., 2017; Adebiyi and Zuidema, 2018).

In the statistical models atmospheric stability, air-mass dynamics and relative humidity at cloud level are the most important

15 drivers for changes in CF and REF, relative to the given set of predictors. The SEA cloud cover is dominated by LTS in all sub-




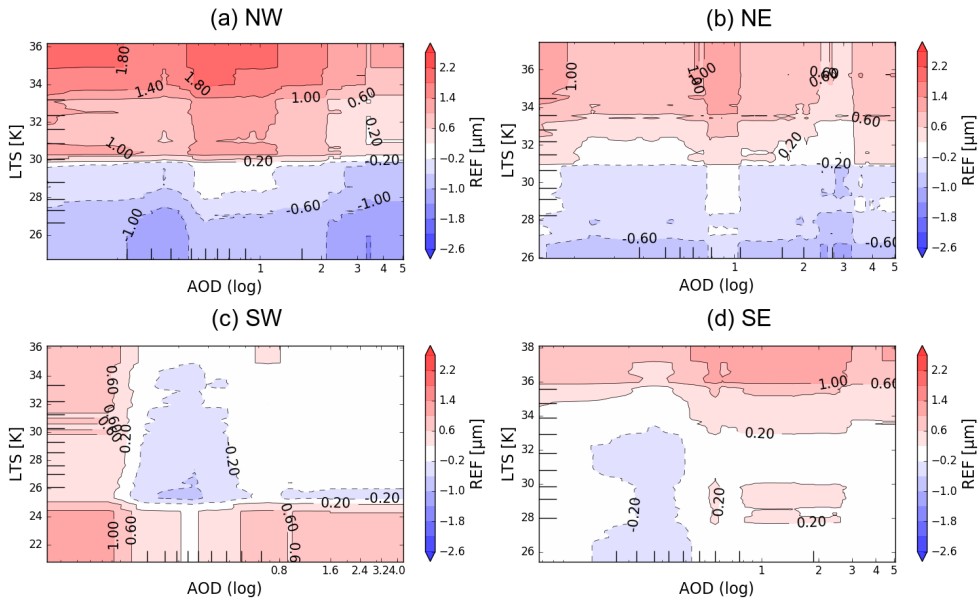

**Figure 8.** Two-variable partial dependence of REF on LTS and AOD in the four subregions NW (a), NE (b), SW (c), SE (d). Description as in Fig. 5.

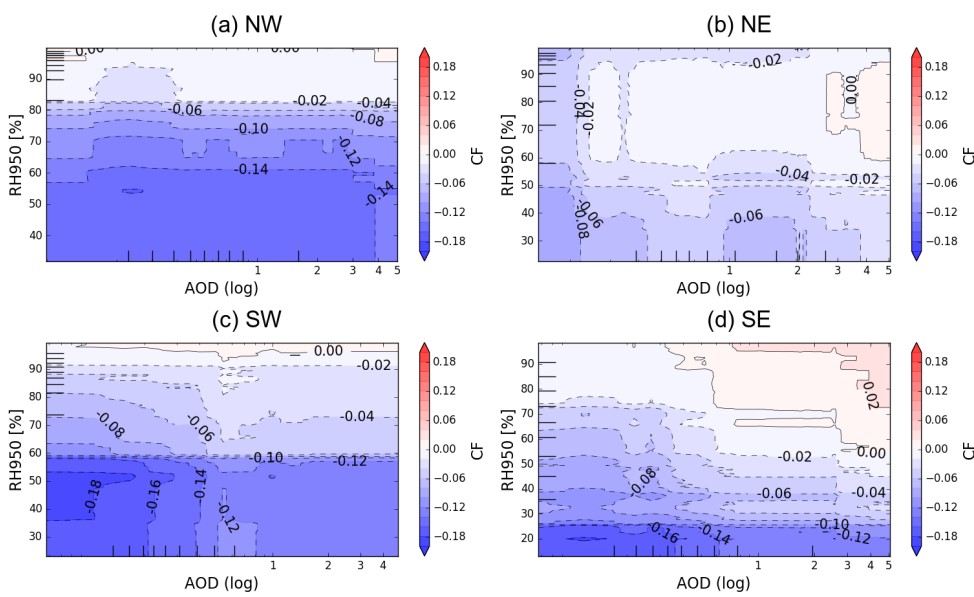

**Figure 9.** Two-variable partial dependence of CF on RH950 and AOD in the four subregions NW (a), NE (b), SW (c), SE (d). Description as in Fig. 5.





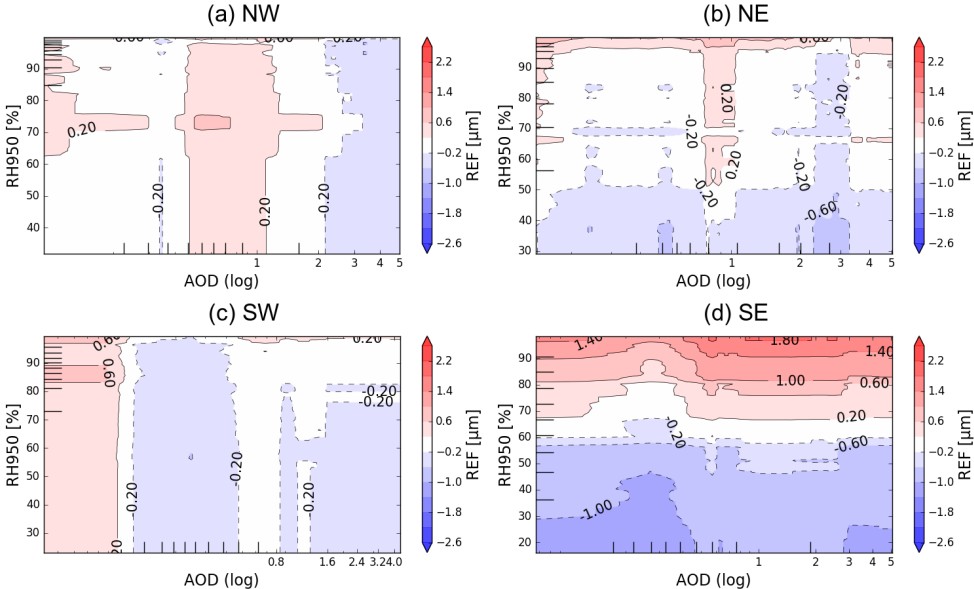

**Figure 10.** Two-variable partial dependence of REF on RH950 and AOD in the four subregions NW (a), NE (b), SW (c), SE (d). Description as in Fig. 5.

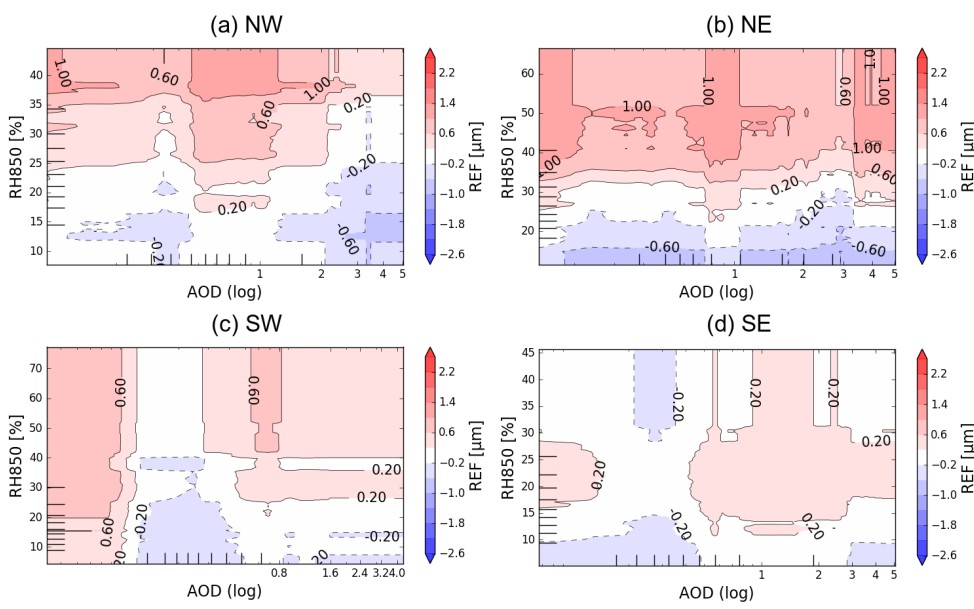

**Figure 11.** Two-variable partial dependence of REF on RH850 and AOD in the four subregions NW, NE, SW, SE. Description as in Fig. 5.

regions. In the NE, cloud amount and droplet size is additionally controlled by surface wind speeds, while in the SE, both are essentially influenced by the availability of moisture. Large-scale dynamics is the main driver of changes of cloud properties





in the SW.

The positive relation between LTS and CF obtained from the GBRT models is explained by the stabilization of the boundary layer dynamics, which promotes cloud amount and longevity. The sensitivity of CF to LTS is non-linear and saturates in stable conditions of LTS > ∼30 K. LTS is especially important in the southern subregions, which are exposed to more variable atmospheric states.

Air-mass dynamics (air-mass origin and zonal wind speeds at 600 hPa) determine REF in the SW to a greater extent than in the NE. The REF increase in the SW is attributed to the outreach of convective westerly disturbances to this subregion. In the NE, air masses show less variability as they approach mainly from the continent under more stable conditions. Here, dynamically induced strong wind speeds and a warm free troposphere are associated with larger droplets.

Although aerosols play a secondary role for the prediction of cloud properties, important implications for the subregional 
strength of ACI can be derived from the model's partial dependencies. In the southern subregions, a strong sensitivity of CF and REF to AOD is modeled, likely due to aerosol-cloud interactions and semi-direct effects. CF sensitivities to aerosols are shown to be stronger in stable conditions, where dry-air entrainment is reduced. A higher REF sensitivity in unstable conditions is attributed to e.g. generally larger droplets, a different aerosol composition (e.g. sea salt) and a more turbulent layer, which possibly favors stronger aerosol indirect effects in these regions. Outcomes also point to the saturation of the aerosol indirect 
effect in the NE compared to the SW where low aerosol loadings may more efficiently act as cloud condensation nuclei.

This study presents the potential of using multivariate GBRTs to derive cloud determinants, and non-linear sensitivities and further to give realistic estimates of the magnitude of aerosol relationships on a synoptic scale. Due to the limited capability of a statistical model to learn the data inherent relations only, feedback mechanisms and satellite artifacts in the SEA cannot completely be accounted for. However, the application of machine learning techniques is advantageous and yields valuable 
insights into subregional cloud and ACI processes on the microphysical and macrophysical scale.

*Competing interests.* The authors declare that they have no conflict of interest.

*Acknowledgements.* MODIS data were obtained from the Goddard Space Flight Center (http://ladsweb.nascom.nasa.gov/data/search.html). The authors gratefully acknowledge the NOAA Air Resources Laboratory (ARL) for the provision of the HYSPLIT transport model (http: 
//ready.arl.noaa.gov/HYSPLIT.php). ERA-Interim data were obtained from the homepage of European Centre for Medium-Range Weather Forecasts (http://apps.ecmwf.int). CALIPSO data were accessed through the NASA Langley Research Center Atmospheric Science Data Center (https://eosweb.larc.nasa.gov). The contribution of Hendrik Andersen was supported by Deutsche Forschungsgemeinschaft (DFG) in the project Namib Fog Life Cycle Analysis (NaFoLiCA), CE 163/7-1.



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
