# Peer review of "Building a cloud in the Southeast Atlantic: Understanding low-cloud controls based on satellite observations with machine learning"

_Atmospheric Chemistry and Physics, 2018_

## Referee Comment (RC1) · Anonymous Referee #1 · 3 Aug 2018

"Building a cloud in the Southeast Atlantic: Understanding low-cloud controls based on satellite observations with machine learning" by Fuchs et al. applies a machine-learning program to satellite observations and studies the factors that influence cloud properties in the southeast Atlantic. The method is novel and by itself worthy of publication. The findings on sub-regional variability in dominant factors are interesting and promote better understanding of the climate in the region. The manuscript is written well. I recommend publication. The authors may consider the following suggestions.

Discussion on the data size and the robustness of statistics would be helpful. The variables and their spatial and temporal ranges are given in Section 2.1 and Section

[Figure]
**Interactive comment**

2.2. But I find it difficult to determine whether some sharp features (e.g., in Figure 3d around 282.7K) are a result of poor counting statistics.

Page 1, line 20. Remove the first comma.

Page 3, line 33. What is meant by "generalize, its performance and computational demand"?

Page 5, line 12. Rephrase "relative humidity is essential for cloud formation processes and characteristics".

Page 6, line 15. Break down the long sentence.

Page 7, line 8. "LTS is most sensitive to CF". Did you mean "CF is most sensitive to LTS"?

Page 10, line 10. "the reduction of CF by subsiding dry air". Isn't subsidence usually associated with higher stability and more clouds?

Page 10, the paragraph starting in line 27, or later. Figures 5-11 are from only one model run selected at random. How representative are these snapshots of all model runs?

Page 11, line 1. Remove the first comma.

Page 11, line 3. Remove the first comma.

---

## Referee Comment (RC2) · Anonymous Referee #2 · 7 Aug 2018

This manuscript disentagles aerosol effects on the southeast Atlantic stratocumuls deck from meteorological effects through the use of a machine learning approach labeled Gradient Boosting Regression Trees (GBRTs). It is welcome to see a recognition of both impacts, and the use of an innovate approach to discriminate them. The use of lat_src and lon_src is nice. The results are sensible. I do however feel the study suffers from over-interpretation. One concern is the focus on only the cloud fraction and the cloud effective radius (REF) as the cloud properties. While the REF is influenced by aerosol, it is also a function of the liquid water path. A more straightforward physical relationship is that between AOD (CCN) and the cloud droplet number concentration (Nd), which can be estimated as a function of REF and the cloud optical depth. Cloud

deepening is likewise better interpreted through the use of LWP than of REF. Another concern is the lumping of July-August-September. It is by now well appreciated that the biomass-burning aerosol is more likely to be present within the boundary layer in July, moving up in altitude through September, when it is more likely to be above the cloudy boundary layer. Different cloud responses would be anticipated as a function of the month. A useful additional analysis is to examine the GBRT results as a function of month, and interpret them as a function of the varying cloud-aerosol vertical structure.

Other comments follow:

1. I am not completely comfortable with the use of the 8-day MODIS L3 product used as opposed to shorter time scale, as the 8-day time scale will average over the synoptic time scale and is far longer than the cloud adjustment time scale of 1-2 days. The authors mention that an 8-day time scale "allows for the large-scale and thermodynamic forcings of cloud properties to be combined", but I remain unclear what this means exactly. In several places in the manuscript the authors refer to processes that occur at much smaller time scales, such as the cloud microphysical response to aerosol. Instead it seems to me the 8-day time scale is primarily capturing a portion of the monthly evolution in the aerosol-cloud vertical structure and seasonal meteorological cycle. Also, the 8-day time scale should be explicitly mentioned in the abstract. 2. an issue with using the relative humidity at 950 hPa is that changes in RH are more likely to reflect co-variations with other factors such as the cold-temperature advection (I suspect this explains the stronger relationship between RH_950hpa and REF in the SE sub-region) and cloud-top inversion strength. Have the authors examined the cross-correlations between their predictors? 3. how is it that the machine learning approach is able to grasp non-linear relationships? The description of the technique presented on p. 4 still seems to present it as a basically linear technique. 4. It is worth mentioning that the larger region encompassing the 4 subregions has been previously examined in Klein and Hartmann 1993.

[Figure]

2018.

---

## Author Comment (AC1) · 18 Sep 2018

**Response to interactive comment of anonymous referee 1 —**

Julia Fuchs[1, 2], Jan Cermak[1, 2], and Hendrik Andersen[1, 2]

[1]Institute of Meteorology and Climate Research, Karlsruhe Institute of Technology (KIT), Karlsruhe, Germany.

[2]Institute of Photogrammetry and Remote Sensing, Karlsruhe Institute of Technology (KIT), Karlsruhe, Germany.

contact: julia.fuchs@kit.edu

"Building a cloud in the Southeast Atlantic: Understanding low-cloud controls based on satellite observations with machine learning" by Fuchs et al. applies a machine- learning program to satellite observations and studies the factors that influence cloud properties in the southeast Atlantic. The method is novel and by itself worthy of publication. The findings on sub-regional variability in dominant factors are interesting and promote better understanding of the climate in the region. The manuscript is written well. I recommend publication. The authors may consider the following suggestions.

**General Comments:**

Discussion on the data size and the robustness of statistics would be helpful. The variables and their spatial and temporal ranges are given in Section 2.1 and Section 2.2. But I find it difficult to determine whether some sharp features (e.g., in Figure 3d around 282.7K) are a result of poor counting statistics.

The GBRT models are computed based on approximately 2000 data points per parameter (now added on p.3, l.19). A robust performance of these models is shown in terms of the R2 (NRMSE), which presents a good agreement of the predicted vs. observed cloud property based on 10 model runs of an independent (unseen) dataset. The robustness of the model toward overfitting to the training dataset is ensured by the cross-validated tuning of the hyperparameter, the choice of the robust Huber loss function and the implementation of an early stopping rule. The section of the manuscript (p.4, l.9) is modified for clarity.

The sharp feature observed in Figure 3d for the NE subregion is shown in Fig. 1 together with a 2-dimensional frequency plot of the total data counts and the data mean per T700 bin. A good agreement between modeled and observed relationship is shown, and the sharp feature is associated with sufficient data. Thus, this case shows how the model is able to capture the data inherent

relationships. However, the marked steps in the partial dependencies (e.g. Fig. 5) are most likely artifacts due to the decision tree based algorithm. This aspect is added on p.5, l.2: "Marked steps in the partial dependencies have to be interpreted with caution (e.g. Fig. 5), as they can be in part caused by the decision tree based algorithm, dividing the parameter space into separate regions.".

[Figure]

Figure 1: Predicted (GBRTs; red) versus observed (Obs.; black) mean REF binned to 98 T700 percentiles (1st - 99th) of the observation data for the NE subregion. Two-dimensional absolute frequencies of observations colored in blue.

**Detailed Comments:**

Page 1, line 20. Remove the first comma.

Done.

Page 3, line 33. What is meant by "generalize, its performance and computational demand"?

The ability of a GBRT model to generalize means that the model is capable to predict an output with good agreement to the observations (R2, NRMSE) based on an unseen dataset. The more the model learns (without overfitting to the dataset) the better it is able to predict (performance), however, the longer is the training and running time for the model to be computed. The sentence is rephrased as follows: "In general, a high number of boosting iterations and a low learning rate will increase the models ability to make predictions on an unseen dataset (generalize), its performance and computational demand during training."(p.4, l.1)

Page 5, line 12. Rephrase "relative humidity is essential for cloud formation processes and characteristics".

The sentence is rephrased as follows: "As free tropospheric and cloud-level humidity influence dry-air entrainment and cloud characteristics in marine low clouds (Wood, 2012; Jones et al., 2014; Bretherton et al., 2013; Andersen et al., 2017), relative humidity values at 700, 850 and 950 hPa are selected as predictors." (p.5, l.21)

Page 6, line 15. Break down the long sentence.

The sentence is broken down as follows: "The application of the GBRTs aims at finding subregional patterns of relevant low-cloud drivers, without creating a model which fully covers the interactions between clouds and their environmental conditions. The predictor set was selected in a way to reduce covariation. Thus, the choice of predictors reflects the compromise between characterizing the atmospheric state sufficiently without creating a model that lacks interpretability."(p.6, l.25)

Page 7, line 8. "LTS is most sensitive to CF". Did you mean "CF is most sensitive to LTS"?

Yes, thanks for this comment. The sentence is modified accordingly.

Page 10, line 10. "the reduction of CF by subsiding dry air". Isn't subsidence usually associated with higher stability and more clouds?

Yes, however, a study by Myers and Norris (2013) showed further that subsidence can also reduce cloudiness for the same value of LTS, which is explained by a lowering of the marine boundary layer. The reference is added to the manuscript (p.9, l.32).

Page 10, the paragraph starting in line 27, or later. Figures 5-11 are from only one model run selected at random. How representative are these snapshots of all model runs?

The two-variable partial dependencies are essentially the same as the one-variable partial dependencies only for two predictors. Thus, the same range between maximum and minimum of the one-variable partial dependence obtained from all model runs (shaded area in e.g. Fig. 3) is expected for the two-variable partial dependencies. This is now mentioned in the caption of Fig. 5: "For this illustration only one model run is selected at random as it represents all model runs with error ranges comparable to that of the one-variable partial dependencies." Figure 2 shows similar patterns obtained from three different model runs.

[Figure]

Figure 2: Two-variable partial dependence of REF on Lon_src and Lat_src in the in the SW subregion. The three panels show three SW model runs selected at random.

Page 11, line 1. Remove the first comma.

Done.

Page 11, line 3. Remove the first comma.

Done.

**References**

Andersen, H., Cermak, J., Fuchs, J., Knutti, R., and Lohmann, U. (2017). Understanding the drivers of marine liquid-water cloud occurrence and properties with global observations using neural networks. *Atmospheric Chemistry and Physics*, 17(15):9535–9546.

Bretherton, C. S., Blossey, P. N., and Jones, C. R. (2013). Mechanisms of marine low cloud sensitivity to idealized climate perturbations: A single-LES exploration extending the CGILS cases. *Journal of Advances in Modeling Earth Systems*, 5(2):316–337.

Jones, C. R., Bretherton, C. S., and Blossey, P. N. (2014). Fast stratocumulus time scale in mixed layer model and large eddy simulation. *Journal of Advances in Modeling Earth Systems*, (6):206–222.

Myers, T. A. and Norris, J. R. (2013). Observational evidence that enhanced subsidence reduces subtropical marine boundary layer cloudiness. *Journal of Climate*, 26(19):7507–7524.

Wood, R. (2012). Stratocumulus Clouds. *Monthly Weather Review*, 140(8):2373–2423.

---

## Author Comment (AC2) · 18 Sep 2018

**Response to interactive comment of anonymous referee 2 —**

Julia Fuchs[1,2], Jan Cermak[1,2], and Hendrik Andersen[1,2]

[1]Institute of Meteorology and Climate Research, Karlsruhe Institute of Technology (KIT), Karlsruhe, Germany.
[2]Institute of Photogrammetry and Remote Sensing, Karlsruhe Institute of Technology (KIT), Karlsruhe, Germany.

contact: julia.fuchs@kit.edu

This manuscript disentangles aerosol effects on the southeast Atlantic stratocu-muls deck from meteorological effects through the use of a machine learning approach labeled Gradient Boosting Regression Trees (GBRTs). It is welcome to see a recognition of both impacts, and the use of an innovate approach to discriminate them. The use of lat_src and lon_src is nice. The results are sensible. I do however feel the study suffers from over-interpretation. One concern is the focus on only the cloud fraction and the cloud effective radius (REF) as the cloud properties. While the REF is influenced by aerosol, it is also a function of the liquid water path. A more straightforward physical relationship is that between AOD (CCN) and the cloud droplet number concentration (Nd), which can be estimated as a function of REF and the cloud optical depth. Cloud deepening is likewise better interpreted through the use of LWP than of REF. Another concern is the lumping of July-August-September. It is by now well appreciated that the biomass-burning aerosol is more likely to be present within the boundary layer in July, moving up in altitude through September, when it is more likely to be above the cloudy boundary layer. Different cloud responses would be anticipated as a function of the month. A useful additional analysis is to examine the GBRT results as a function of month, and interpret them as a function of the varying cloud-aerosol vertical structure.

The study was designed to focus on cloud fraction and cloud effective radius in order to test and interpret the GBRT models on one relevant micro- and one relevant macrophysical cloud property during the biomass-burning season. Using the cloud droplet number concentration is appreciated, however, as it is derived from COT and REF, and based on assumptions on cloud vertical profile, additional uncertainties would be introduced (Grosvenor et al., 2018). We chose to avoid this, because we try to capture the cloud system as completely as possible with the statistical model. As such, we include information on factors that also determine LWP. As variability among these LWP-predictors is simulated in the computation of the sensitivities, we thereby indirectly con- strain LWP effects on REF. To account for the referee's suggestion LWP as an essential cloud property is analyzed and results support the interpretation of cloud thickening under stable conditions in all subregions (Fig. 1a), as well as during westerly disturbances, especially in the western subregions (Fig. 1b).

In the manuscript the LWP-effects refer to outcomes of a comparable study by

Fuchs et al. (2017) where LWP is discussed and 'self-constraining model' is now detailed more clearly.

[Figure]

Figure 1: Mean partial dependence of LWP on LTS (left) and source latitude of air mass (right) in the four subregions (colors).

The aggregation of the months July-August-September (JAS) was conducted for better comparability to previous studies (Painemal et al., 2014; Andersen and Cermak, 2015; Adebiyi and Zuidema, 2018) investigating the same season.

While we agree with referee 2, changes of the aerosol and boundary layer occur on all scales, so that the assumptions outlined by referee 2 need to be made independent of scale. The intraseasonal variability contained in the training of the GBRT model contribute to the relationships during the investigated season and must be taken into account for the interpretation of results. For this reason, we have now computed monthly GBRT models and included the results concerning the aerosol-cloud relationships in the manuscript. The following figure and text are added to the manuscript. "Figure 7 shows AOD-REF

[Figure]

Figure 2: Mean partial dependence of REF on AOD in the four subregions (colors) in July (a) and September (b).

partial dependencies for the months of July and September separately. While during July, REF seems to decrease with increasing AOD, especially in the SW subregion, during September the opposite relationship is found. The contrasting relationships may be related to differences in the vertical distribution of aerosols and clouds in the Southeast Atlantic. During July, aerosol and cloud layers are frequently entangled, facilitating ACI, whereas in September they can be well separated (Adebiyi et al. 2018). During this time, absorbing aerosol may increase the stability and trap humidity in the boundary layer, potentially leading to the observed relationship. The JAS partial dependence between AOD and REF can thus be viewed as a summary of these patterns. However, it is not the study's focus to separate the different aerosol effects mentioned earlier, but to analyze the overall influence of aerosols on clouds during the biomass-burning season."(p.14, l.10)

**Other comments follow:**

1. I am not completely comfortable with the use of the 8-day MODIS L3 product used as opposed to shorter time scale, as the 8-day time scale will average over the synoptic time scale and is far longer than the cloud adjustment time scale of 1-2 days. The authors mention that an 8-day time scale "allows for the large-scale and thermodynamic forcings of cloud properties to be combined", but I remain unclear what this means exactly. In several places in the manuscript the authors refer to processes that occur at much smaller time scales, such as the cloud microphysical response to aerosol. Instead it seems to me the 8-day time scale is primarily capturing a portion of the monthly evolution in the aerosol-cloud vertical structure and seasonal meteorological cycle. Also, the 8-day time scale should be explicitly mentioned in the abstract. The study focuses on processes on aggregated time scales (8-day), assuming that cloud adjustments due to aerosols, though acting on smaller time scales, are detectable in the aggregated data set at the same time as changes of thermodynamic and dynamic conditions. While daily or hourly data might underestimate e.g. the effect of LTS on the cloud cover, the influence of aerosols on the cloud cover might be underestimated by the 8-day aggregation. In particular, since aerosol and cloud properties are not retrieved at the same time in a given location. Eight-day averages are taken to represent the mean states of both at that time scale. These aspects are important and now more explicitly addressed in the manuscript. "The temporal resolution of 8 days allows to combine large-scale, thermodynamic and aerosol forcings of cloud properties simultaneously on a synoptical scale. However, it must be taken into account that clouds adjust on different time scales (hours to several days) to their environment (Klein, 1997; McCoy et al., 2017; Adebiyi and Zuidema, 2018) and thus processes relevant on shorter time scales might be underrepresented in the data set."(p.3, l.8) The 8-day time scale is now introduced in the abstract (p.1, l.5).

2. an issue with using the relative humidity at 950 hPa is that changes in RH are more likely to reflect co-variations with other factors such as the cold-temperature advection (I suspect this explains the stronger relationship between RH_950hpa and REF in the SE sub-region) and cloud-top inversion strength. Have the authors examined the cross-correlations between their predictors?

Thank you for pointing out this issue. Correlations between the predictors were examined in advance and influenced the choice of predictors to reduce the covariation. Cold-temperature advection and cloud-top inversion strength are not explicitly chosen as predictors, but are assumed to be represented in the data set by other parameters such as wind speed, sea surface temperature and LTS. The manuscript is modified on p.6, l.26: "The predictor set was selected in a way to reduce covariation."

3. how is it that the machine learning approach is able to grasp non-linear relationships? The description of the technique presented on p. 4 still seems to present it as a basically linear technique.

The GBRT algorithm is based on decision trees which are capable of representing non-linear dependencies between predictor and predictand. The parameter space is iteratively split with the goal to minimize a loss function. The sum of the linear decisions of each tree in the ensemble can then represent non-linear relationships. The manuscript is modified on p.3, l.24.

4. It is worth mentioning that the larger region encompassing the 4 subregions has been previously examined in Klein and Hartmann 1993.

This reference is added on p. 3, l.13: "In this study CF and REF are simulated based on a selected predictor set (AOD and meteorological parameters) in the SEA ($10°$–$20°$ S, $0°$–$10°$ E, as analyzed in Klein and Hartmann (1993)) using Gradient Boosting Regression Trees (GBRTs)."

**References**

Adebiyi, A. A. and Zuidema, P. (2018). Low Cloud Cover Sensitivity to Biomass-Burning Aerosols and Meteorology over the Southeast Atlantic. *Journal of Climate*, 31(11):4329–4346.

Andersen, H. and Cermak, J. (2015). How thermodynamic environments control stratocumulus microphysics and interactions with aerosols. *Environmental Research Letters*, 10(2):024004.

Fuchs, J., Cermak, J., Andersen, H., Hollmann, R., and Schwarz, K. (2017). On the Influence of Air Mass Origin on Low-Cloud Properties in the Southeast Atlantic. *Journal of Geophysical Research: Atmospheres*, 122(20):11,076–11,091.

Grosvenor, D. P., Sourdeval, O., Zuidema, P., Ackerman, A., Alexandrov, M. D., Bennartz, R., Boers, R., Cairns, B., Chiu, J. C., Christensen, M., Deneke, H. M., Diamond, M. S., Feingold, G., Fridlind, A., Hünerbein, A., Knist, C. L., Kollias, P., Marshak, A., McCoy, D., Merk, D., Painemal, D., Rausch, J., Rosenfeld, D., Russchenberg, H., Seifert, P., Sinclair, K., Stier, P., van Diedenhoven, B., Wendisch, M., Werner, F., Wood, R., Zhang, Z., and Quaas, J. (2018). Remote sensing of droplet number concentration in warm clouds: A review of the current state of knowledge and perspectives. *Reviews of Geophysics*.

Klein, S. and Hartmann, D. (1993). The seasonal cycle of low stratiform clouds. *Journal of Climate*, 6:1587–1606.

Klein, S. A. (1997). Synoptic Variability of Low-Cloud Properties and Meteorological Parameters in the Subtropical Trade Wind Boundary Layer. *Journal of Climate*, 10(8):2018–2039.

McCoy, D. T., Eastman, R., Hartmann, D. L., and Wood, R. (2017). The change in low cloud cover in a warmed climate inferred from AIRS, MODIS, and ERA-interim. *Journal of Climate*, 30(10):3609–3620.

Painemal, D., Kato, S., and Minnis, P. (2014). Boundary layer regulation in the southeast Atlantic cloud microphysics during the biomass burning season as seen by the A-train satellite constellation. *Journal of Geophysical Research:*

*Atmospheres*, 119(19):11,288–11,302.

---

## Author Response (AR2)

**Response to the editor —**

Julia Fuchs[1, 2], Jan Cermak[1, 2], and Hendrik Andersen[1, 2]

[1]Institute of Meteorology and Climate Research, Karlsruhe Institute of Technology (KIT),
Karlsruhe, Germany.
[2]Institute of Photogrammetry and Remote Sensing, Karlsruhe Institute of Technology
(KIT), Karlsruhe, Germany.

contact: julia.fuchs@kit.edu

Co-Editor Decision: Publish subject to minor revisions (review by editor) (06 Oct 2018) by Timothy J. Dunkerton

**Comments to the Author:**

Please clarify a few things here:

"The ERA-Interim reanalysis data is also used in the calculation of 5-days backward air-mass trajectories with the HYSPLIT model using geopotential height, relative humidity, temperature, u/v wind components, vertical velocity at 37 pressure levels. The backward trajectories 5 are initialized at 12 UTC, at each grid point of the study area and at a subregional mean cloud-top altitude obtained from the CALIPSO Level-2 5 km layer cloud product (version 3, daytime). All meteorological variables are resampled from 0.5 degrees to the MODIS L3 resolution of 1 degrees and temporally averaged to the corresponding 8-day product."

1. The ERA-Interim data are provided at 0.7 deg horizontal resolution, if it is the same dataset that I use in research.

The ERA-Interim data are interpolated to 0.5 degrees for the trajectory analysis and resampled to the MODIS L3 resolution of 1 degree. The use of 0.5 instead of the native 0.75 degree data is not expected to affect the data accuracy.

2. The 37 pressure levels extend to 1 hPa (stratopause), presumably a small subset of these are used for trajectory analysis.

Yes, thanks. The trajectory modeling is based on a subset of 25 pressure levels, with the highest trajectory reaching approximately 4 km. The manuscript is modified.

3. Presumably the trajectories are driven by 6-hourly analyses or perhaps

interpolated to something finer in time, in any case, they are certainly not driven by 8-day averaged data.

The trajectories are driven by 6-hourly analyses. This is now clarified in the manuscript.

A general suggestion: If possible, please identify supporting literature that would justify the use of 8-day meteorological averages. While cloud properties are justifiably driven by variations on all timescales, they also exhibit virtually instantaneous response to changes, e.g, near water-phase transitions!

We agree with the editor. The cloud response may vary from 6 to 12 hours for water path adjustments until several days for cloud deepening (Mauger and Norris, 2010; Jones et al., 2014). The latter effect would be underrepresented in a daily dataset. The 8-day averages are used in McCoy et al. (2017) as well. Publications are added to the manuscript.

**References**

[revised manuscript text omitted]